

# Femoral metadiaphyseal and nutrient foramen perfusion suggests comparable maximal metabolic rates in a pterosaur and in a semi-aquatic maniraptoran dinosaur

Mariana V.A. Sena[1], Dennis F.A.E. Voeten[2], Esaú Araújo[3] and Jorge Cubo[1]

[1] Sorbonne Université, Muséum National d'Histoire Naturelle, CNRS, Centre de Recherche en Paléontologie–Paris (CR2P, UMR 7207), Paris, France
[2] Frisian Museum of Natural History, Leeuwarden, Netherlands
[3] Museu Nacional, Universidade Federal do Rio de Janeiro, Rio de Janeiro, Brazil

## ABSTRACT

Ornithodirans represent a diverse and highly successful clade that encompasses a wide array of morphologies and ecological adaptations. This group includes volant forms such as *Rhamphorhynchus*, a medium-sized, non-pterodactyloid long-tailed pterosaur from the Jurassic Solnhofen lagoons, characterized by prow-shaped lower jaw and forward-pointing teeth consistent with a piscivorous diet. In addition, the ornithodiran group included theropod dinosaurs such as *Halszkaraptor escuilliei*, a dromaeosaurid from Mongolia that exhibit morphological traits indicative of a semi-aquatic lifestyle. In this study, we retrodicted the aerobic performance of these two extinct ornithodirans by using femoral metadiaphyseal nutrient foramina dimensions as a proxy for maximal metabolic rate (MMR), based on an extant-amniote dataset of reference. We estimated femoral blood flow ($\dot{Q}$) through the femoral nutrient canal areas and retrodicted mass-independent MMR for immature specimens of *Halszkaraptor* and *Rhamphorhynchus* of 5.68 mLO2 h$^{-1}$ g$^{-0.87}$ and 5.55 mLO2 h$^{-1}$ g$^{-0.87}$ ($p < 0.001$), respectively. Our findings revealed that femoral blood flow rates and mass-independent MMR values were similar in the two taxa, despite their extreme differences in phylogenetic affinity, locomotory behavior and ecology. The predicted mass-independent MMR for *H. escuilliei* aligned with values observed in extant ground-dwelling emus and migratory shorebirds such as gulls and terns, but fall below the high MMR value of ducks. Further investigation into adult specimens is needed to refine our understanding of aerobic capacity in mature individuals, particularly with regards to the ability of *Rhamphorhynchus* for achieving the energetic demands of flight. Our research enhances understanding of the physiological strategies of extinct taxa and helps address key gaps in paleophysiological reconstructions.

Corresponding author
Mariana V.A. Sena,
mari.araujo.sena@gmail.com

## INTRODUCTION

Maximal metabolic rate (MMR), also referred to as the maximal rate of oxygen consumption ($VO_2$max), represents the highest aerobic metabolic rate achieved during exercise and defines the upper limit of sustained metabolic performance (*Bennett & Ruben, 1979*; *Weibel et al., 2004*). In extant amniotes, MMR can be measured directly using a circuit respirometer during physical activity. For volant taxa, this involves measurements during forced flight (*Hails, 1979*; *Norberg, 1996*), while for terrestrial species, metabolic rates are assessed while running on a treadmill (*Fedak, Pinshow & Schmidt-Nielsen, 1974*; *Taylor, Heglund & Maloiy, 1982*; *Seymour, Runciman & Baudinette, 2008*). For swimming amniotes, such as ducks, penguins and sea turtles, MMR is measured in controlled-flow swim channels (*Prange & Schmidt-Nielsen, 1970*; *Prange, 1976*; *Kooyman & Ponganis, 1994*).

Reconstructing the maximal aerobic capacity of extinct archosaurs has posed significant challenges, leading to the development of an indirect method, referred to as the "foramen technique"; developed by *Seymour et al. (2012)*, this technique has proven to be particularly effective (*Knaus et al., 2021*; *Varela, Tambusso & Fariña, 2024*). This approach estimates regional blood flow by quantifying the cross-sectional aperture area of vascular foramina in long bones. Long bones require blood perfusion for remodeling to *e.g.*, repair microfractures induced by locomotion and weight-bearing stresses (*Lieberman et al., 2003*; *Eriksen, 2010*). Nutrient foramen size provides a useful proxy for bone perfusion and aerobic capacity (*Seymour et al., 2012*). A nutrient artery enters the femoral shaft through the nutrient foramen, typically accompanied by a vein (*Currey, 2002*). The size of the foramen correlates dynamically with the size of the vessels it contains. Thus, the MMR scales with the size of the nutrient artery and the corresponding blood flow supplying the bone (*Seymour et al., 2019*). In mature terrestrial vertebrates, femoral blood flow is known to be positively correlated with locomotor activity, with relatively larger nutrient foramina being detected in species with elevated metabolic rates during locomotion (*Seymour et al., 2012*; *Allan et al., 2014*; *Newham et al., 2020*). According to *Seymour et al. (2012)*, the proximate causation of this relationship would be linked to the fact that tetrapods with high levels of activity encounter biomechanical constraints invoking microfractures that are repaired by secondary bone remodeling, thus necessitating high levels of oxygen consumption. The application of phylogenetic eigenvector maps (PEMs; *Guénard, Legendre & Peres-Neto, 2013*) to reconstruct metabolic rates in extinct amniotes has been adopted over the last decade (*Legendre et al., 2016*; *Fleischle, Wintrich & Sander, 2018*; *Cubo et al., 2024*). PEMs provide a framework to infer the metabolic profiles of extinct organisms by integrating phylogenetic and morphological data. In combination with the foramen technique, PEMs have been used to estimate mass-independent MMR in fossil archosaurs, such as extinct pseudosuchians (*Cubo et al., 2024*; *Sena et al., 2023*; *Sena et al., 2025*).

In this study, we used these methods by integrating a comprehensive phylogenetic amniote database with their corresponding nutrient artery blood flow measurements computed using the foramen technique and previously measured MMR values. This combined approach was applied to retrodict the mass-independent MMR of two

extinct ornithodirans with extremely different lifestyles and locomotory strategies: *Rhamphorhynchus* and *Halszkaraptor escuilliei*. *Rhamphorhynchus* was a medium-sized, piscivorous pterosaur from the Late Jurassic Plattenkalks of southern Germany. Characterized by a prow-shaped lower jaw and forward-angled (procumbent) teeth, *Rhamphorhynchus* was adapted to forage extensively in aquatic environments (*Voeten et al., 2018*; *Witton, 2018*). *H. escuilliei* was late Cretaceous dromaeosaurid theropod hypothesized to possess an amphibious eco-morphology, potentially relying on neck hyper-elongation for predatory foraging (*Cau et al., 2017*; *Cau, 2020*). In this research, we aimed to elucidate the metabolic capacities of these two locomotory diverse ornithodirans.

## MATERIALS & METHODS

### Material and foramen measurements

We examined three-dimensional (3D) models of the pterosaur *Rhamphorhynchus* sp. (Musée des Confluences, Lyon, France [MdC]: 20269891) from Solnhofen Plattenkalk, Germany (Upper Jurassic) and the theropod *Halskaraptor escuilliei* (Institute of Paleontology and Geology, Mongolian Academy of Sciences, Ulaanbaatar, Mongolia [MPC]: D-102/109) from the Djadokhta Formation, Mongolia (Campanian). These fossils were imaged using propagation phase-contrast synchrotron microtomography at the European Synchrotron Radiation Facility, Grenoble, France (ESRF), for previous microanatomical and morphological investigations (*Cau et al., 2017*; *Voeten et al., 2018*).

Metadiaphyseal and nutrient foramina openings were measured from three-dimensional (3D) reconstructions of *H. escuilliei* and *Rhamphorhynchus* sp. The foraminal aperture areas of the femora were measured with ImageJ/Fiji (*Schindelin et al., 2012*) (https://imagej.net/software/fiji/). For comparative purposes, we assumed that the foramen aperture area was the same as that of a geometric circle. To estimate nutrient artery blood flow rate ($\dot{Q}$; ml s$^{-1}$) in the femora, we applied the curved polynomial equation: $\log \dot{Q} = -0.20 \log r_i^2 + 1.91 \log r_i + 1.82$ (*Seymour et al., 2019*), where $r_i$ represents the arterial lumen radius extracted from the arterial lumen area which corresponds to approximately 20% of the total foramen area (*Hu, Nelson & Seymour, 2022*).

### MMR retrodictions and recovered unit

To retrodict the mass-independent MMRs of the fossil taxa, we constructed a phylogenetic inference model adapted from the Rscript reported by *Legendre et al. (2016)* using a dataset of extant species collated from published literature composed of MMRs, body masses and the nutrient artery $\dot{Q}$ values of 43 species of amniotes, spanning mammals ($n = 15$), non-avian sauropsids ($n = 14$), and avian sauropsids ($n = 14$) (Table S1). It is well established that larger animals have greater metabolic needs in mLO$_2$ h$^{-1}$ but lower metabolic needs when expressed in mLO$_2$ h$^{-1}$ g$^{-1}$. The metabolic rate in mLO$_2$ h$^{-1}$ is known to increase with body mass following an exponential relationship, with an exponent <1 (*Schmidt-Nielsen, 1984*). We compared three taxonomic groups based on their metabolic features, controlling for mass exponents and evolutionary divergence timing. We adjusted their variable allometric exponents to a common value. We applied an allometric exponent of 0.87 to body mass (in grams) across our amniote set, based on

the phylogenetic mean recovered using the 'Reconstruct ancestral states' function with the 'Character history source' option in the 'Trace' menu of Mesquite (*Maddison & Maddison, 2014*).

This generated a single character matrix of distinct extant species body mass allometric exponents of 0.829 for non-avian sauropsids (*Seymour, 2013*), 0.87 for synapsids (*White & Seymour, 2005*) and 1.02 for avians (*Allan et al., 2014*). However, no body mass allometric exponents were yielded for fossil taxa. This approach is more aligned with evolutionary principles and was taken into adopted in an earlier study (*Sena et al., 2025*). In the present study, the phylogenetic exponent recovered was 0.87; therefore, mass-independent MMR data are provided in units of $mLO_2$ $h^{-1}$ $g^{-0.87}$.

## Phylogenetic framework

We adopted the phylogenetic relationships of Neornithes by *Stiller et al. (2024)*, those of Cetartiodactyla by *Zurano et al. (2019)*, those of other mammals by *Upham, Esselstyn & Jetz (2019)*, those of specifically varanid lizards by *Villa et al. (2018)*, and those of squamates in general by *Pyron, Burbrink & Wiens (2013)* and *Vidal & Hedges (2005)*. Branch length data for extant taxa were collated from the Time Tree of Life (timetree.org, accessed November 29, 2024), while those for extinct taxa were collated from the Paleobiology Database (paleobiodb.org, accessed November 29, 2024).

## Phylogenetic comparative method

Paleobiological inference models for mass-independent MMR were constructed using phylogenetic eigenvector maps (PEM) from the "MPSEM" package (*Guénard, Legendre & Peres-Neto, 2013*) in R (*R Core Team, 2023*). The model allowed us to retrodict mass-independent MMR values, along with their 95% confidence intervals, for the two fossil ornithodirans. The normality of the residuals was checked using the Shapiro–Wilk test.

## RESULTS

Based on their metadiaphyseal and nutrient foramen dimensions, the pterosaur *Rhamphorhynchus* sp. (MdC 20269891) and the theropod *H. escuilliei* (MPC D-102/109) had comparable calculated nutrient artery flow rates ($\dot{Q}$) of 0.0017 ml s$^{-1}$ and 0.0010 ml s$^{-1}$, respectively (Fig. 1).

The retrodicted mass-independent MMR for the "long-tailed" pterosaur *Rhamphorhynchus* sp. is 5.55 $mLO_2$ $h^{-1}$ $g^{-0.87}$ (95% confidence interval CI [4.37–7.05] $mLO_2$ $h^{-1}$ $g^{-0.87}$) while that for the theropod *H. escuilliei* (MPC D-102/109) is 5.68 $mLO_2$ $h^{-1}$ $g^{-0.87}$ (95% CI [4.44–7.26] $mLO_2$ $h^{-1}$ $g^{-0.87}$). The predictive model included the phylogenetic eigenvectors 1, 2, 3, 4, 5, 6, 8, 9, 20, 22, 23, 27, 28, and 41, and the estimated nutrient artery $\dot{Q}$ as the co-predictor (adjusted $R^2 = 0.90$; Akaike Information Criterion corrected (AICc) $= 56.79$; and $p = 3.501e^{-12}$) (Fig. 2). The leave-one-out cross-validation test identified no significant difference between the predicted and observed values for the extant species ($p = 0.7788$).
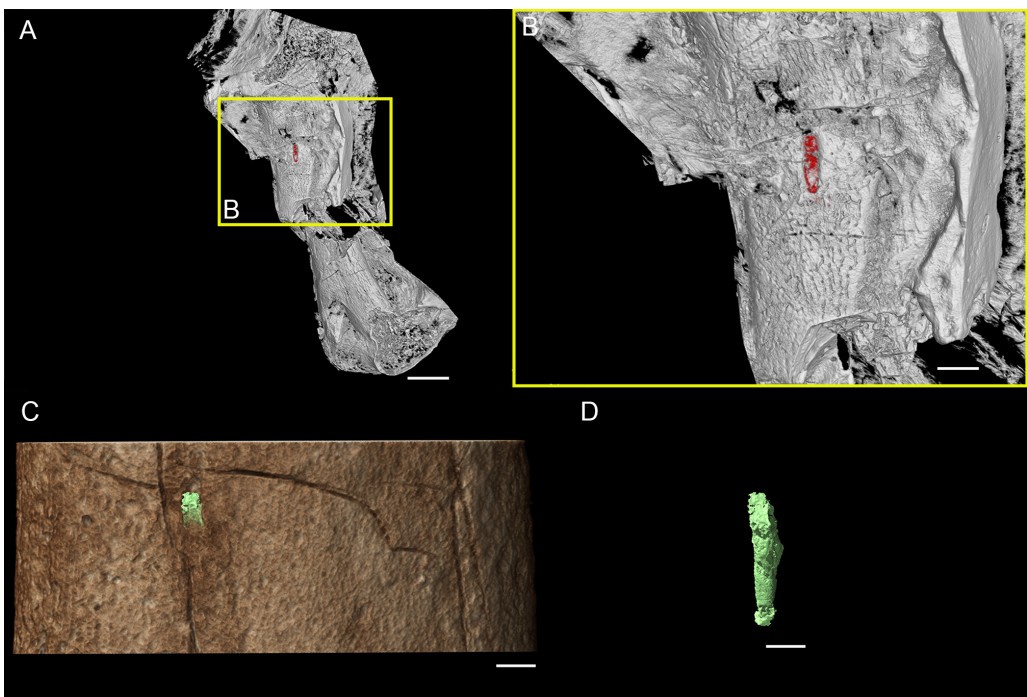

**Figure 1** **Three-dimensional (3D) reconstruction of the femoral nutrient foramina using synchrotron μCT.** *Rhamphorhynchus* sp. (MdC 20269891) femur visualised from tomographic data with a voxel size of 4.24 μm (A) and *Halszkaraptor escuilliei* (MPC D-102/109) mid-diaphyseal femoral interval visualised from tomographic data with a voxel size of 53.58 μm (C), with the identifiable foramina segmented and highlighted in red (B) and virtually extracted in green (D), respectively. Scale bar equals two mm (A), 600 μm (B), and 550 μm (C and D).

## DISCUSSION

The dromaeosaurid *H. escuilliei* (specimen MPC D-102/109) is considered to be a mallard-sized sub-adult theropod, estimated to have a body mass of 1.5 kg (*Cau et al., 2017*). The *Rhamphorhynchus* specimen analyzed in this study weighed approximately 95 g and was identified as a juvenile based on its small size (*Voeten et al., 2018*) with a wingspan ranging from 45 to 50 cm, based on the work described previously by *Prondvai et al. (2012)*. Despite their distinct eco-morphologies, these two ornithodirans exhibited a similar mass-independent MMR and femoral nutrient artery $\dot{Q}$. These data suggest a strong influence of phylogenetic constraints on the outcomes of predictive energy modeling.

We here adopted the morphofunctional interpretation proposed by *Cau et al. (2017)*, proposing that *Halszkaraptor* engaged in subaqueous foraging behavior. Certain pivotal anatomical adaptations of *Halszkaraptor* are comparable to those of extant sawbills, which are considered their ecological analogs (*Cau, 2020*). However, the ecological interpretation of *Halszkaraptor* remains debated. For instance, *Brownstein (2019)* reinterpreted its morphological features as indicative of a transitional form between non-paravian maniraptorans and more derived dromaeosaurids (but also consider *Cau, 2020*). Furthermore, *Fabbri et al. (2022)* argued that the lack of microanatomical specialization

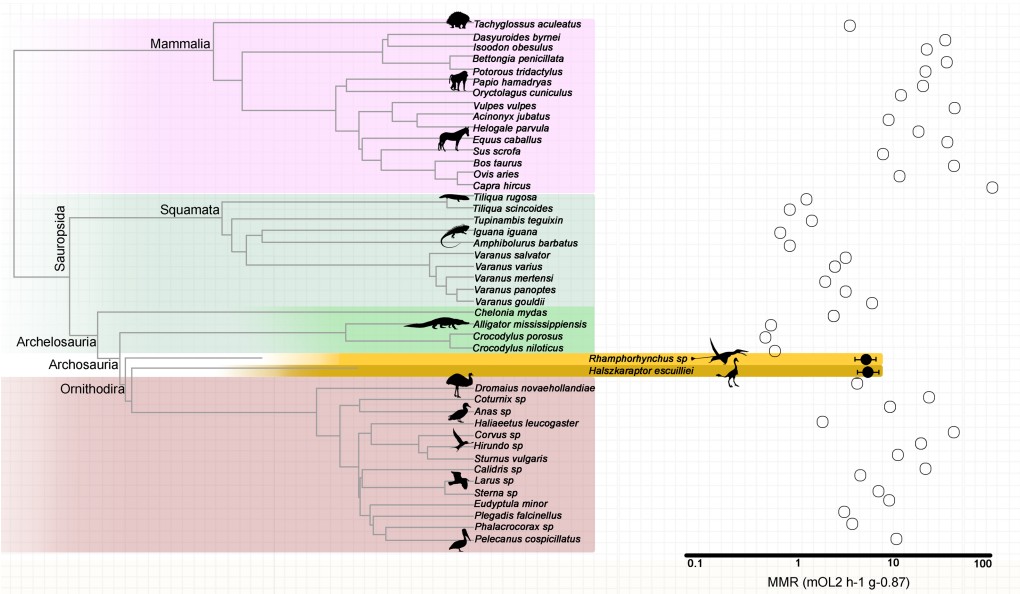

**Figure 2** **Mass-independent maximum metabolic rates of extant amniotes and retrodicted values in fossil archosaurs.** Phylogenetic relationships among extant amniotes informed the mass-independent MMR inference model used to retrodict metabolic rates of the extinct theropod and pterosaur with nutrient artery Q̇ as co-predictor. Branch lengths are proportional to time.

typically associated with semi-aquatic habits may suggest an ambiguous lifestyle for halszkaraptorines. Nevertheless, irrespective of proposed aquatic habits of *Halszkaraptor*, theropods in general (*Legendre et al., 2016*), and the dromaeosaur Velociraptor in particular (*Pontzer, Allen & Hutchinson, 2009*), are generally considered metabolically active taxa, reinforcing *H. escuilliei* as a valid model for suspected elevated dinosaur metabolic capacity.

Our mass-independent MMR estimations for *H. escuilliei* and *Rhamphorhynchus* align with those measured previously in *Varanus gouldii*, a predatory monitor lizard known for its enhanced oxygen transport capabilities, and certain avian taxa, such as emus (*Dromaius novaehollandiae*), gulls (*Larus*), and terns (*Sterna*). Monitor lizards achieve aerobic scopes and sustainable running speeds more than twice those of similar sized, through adaptations, and that include an increased lung surface area, blood buffers, and myoglobin-rich skeletal muscles (*Bartholomew & Tucker, 1964*; *Bennett, 1972*; *Bennett, 1973*). These features underline the importance of morphological investment in oxygen transport systems for enabling high aerobic performance (*Bennett & Ruben, 1979*).

As obligate cursors, flightless ratite emus possess increased hindlimb and heart mass relative to habitual cursorial and incidental burst flying phasianids to facilitate sustained locomotor activity (*Hartman, 1961*; *Grubb, Jorgensen & Conner, 1983*). The VO$_2$max of exercising emus is tenfold higher than their resting values (*Grubb, Jorgensen & Conner, 1983*). Conversely, non-migratory cursorial phasianids, such as guinea fowl (*Numida meleagris*) and jungle fowl (*Gallus*), are adapted for rapid escape flights, maximizing takeoff performance when avoiding predation (*Witter, Cuthill & Bosner, 1994*). These
species exhibit lower endurance and aerobic performance due to the limited aerobic capacity of flight muscle (*Kiessling, 1977*; *Ellerby et al., 2003*; *Askew & Marsh, 2001*; *Henry, Ellerby & Marsh, 2005*). The contribution of flight muscles to organismal aerobic scope in these birds, even when combined with leg activity, is small (*Hammond et al., 2000*). *Halszkaraptor*, interpreted as an amphibious dromaeosaurid, likely combined vigorous hindlimb activity for terrestrial locomotion, similar to flightless ratites, with the use of its forelimbs for swimming. The plesiomorphic glenoid condition characteristic of paravians, which is also inferred for *Halszkaraptor*, may have served as a potential exaptation for a forelimb-assisted swimming style (*Cau et al., 2017*; *Cau, 2020*). However, its aerobic capacity appears to be lower than that of highly specialized swimmers, such as ducks (*Anas* spp.). This suggests that *Halszkaraptor* might be less adapted to sustain longer bouts of swimming compared to mallards.

Rapid growth rates in immature archosaurians have been observed from the Triassic period onward (*Curry Rogers et al., 2024*), which corroborated the active growing of the subadult *Halszkaraptor* individual showing a moderate growth rate (*D'emic et al., 2023*). It also highlights how the generally rapid juvenile growth across archosaurs, as exemplified by *Rhamphorhynchus* MdC 20269891, contributes to overestimation of mass-independent MMR (*Cau et al., 2017*; *Prondvai et al., 2012*; *Araújo et al., 2023*). Previous analyses of *Rhamphorhynchus* suggested that flight capability was likely achieved when individuals reached approximately 30–50% of the adult wingspan and 7–20% of the adult body mass (*Prondvai et al., 2012*). Recent findings indicate that the largest known specimen of *Rhamphorhynchus* had an estimated wingspan of 1.8 m and a body mass of approximately 3 kg (*Hone & McDavid, 2025*). Despite this, evidence of isometric growth and a piscivorous diet in this taxon supports the hypothesis of precocial flight capabilities (*Voeten et al., 2018*; *Hone et al., 2021*). The high mass-independent maximum metabolic rate (MMR) inferred by the present study for our small *Rhamphorhynchus* specimen, representing approximately 3% of the adult body mass, further suggests that powered flight may have been possible even at an early ontogenetic stage. Juveniles often exhibit relatively larger foramen areas due to the increased perfusion required for rapid growth. For example, growing kangaroos possess larger femoral nutrient foramina areas than adults, thus reflecting their higher energy demands for bone development (*Hu et al., 2018*). A similar pattern has been observed in egg-forming oviparous females, in which increased femoral blood flow supports the mobilization of calcium for eggshell formation (*Hu, Nelson & Seymour, 2021*). Thus, the high MMR identified for *Rhamphorhynchus* in this study is more likely associated with its growth demands. Furthermore, this juvenile *Rhamphorhynchus* could have been able to climb trees (*Prondvai et al., 2012*), thus suggesting a high level of ecological activity during early ontogeny, therefore reflecting the energetic demands associated with active exploratory behavior.

As the earliest archosaur group to evolve flapping flight, pterosaurs faced the energetic costs associated with aerial locomotion; this was particularly the case for pterodactyloids with a large body size, such as *Quetzalcoatlus northropi*. The flight energetics of large pterosaurs, such as *Q. northropi*, probably involved anaerobic metabolism during takeoff, followed by energy-efficient soaring or high-speed flight (*Marden, 1994*). To sustain flight

aerobically, *Quetzalcoatlus* likely required an aerobic scope comparable to migratory swans (*Marden, 1994*). In contrast, smaller species of pterosaur, such as *Rhamphorhynchus*, might have had different metabolic requirements during takeoff, highlighting the need for further studies on adult specimens to elucidate their flight endurance and takeoff capabilities.

In volant birds, flying is generally more energetically efficient than running or swimming over equivalent distance due to the higher speeds achieved in flight (*Fedak, Pinshow & Schmidt-Nielsen, 1974*; *Norberg, 1996*; *Butler, 2016*). This aerodynamic efficiency is particularly pronounced in insectivore birds, which exhibit flight costs 49–73% lower than foraging costs measured for non-aerial species (*Hails, 1979*; *Tucker, 1970*). Consequently, the maximal aerobic capacity of adult *Rhamphorhynchus* is expected to be lower than of the juvenile specimen analyzed in this current study.

The mass-independent MMR of juvenile *Rhamphorhynchus* was comparable to that of migratory shorebirds, such as gulls and terns, although despite its elevated aerobic capacity, the juvenile pterosaur exhibits lower mass-specific power output for sustained vertical takeoff when compared to similarly sized migratory phasianids, such as *Coturnix coturnix* and *Coturnix chinensis* (*Bishop, 1997*; *Askew & Marsh, 2001*; *Henry, Ellerby & Marsh, 2005*). Further investigation of adult specimens is required to clarify ontogenetic influence on the flight performance of pterosaurs.

Although *Rhamphorhynchus* and *H. escuilliei* belonged to distinct groups and exhibit different lifestyles, parallels can be drawn with regards to their metabolism and energy efficiency in locomotion. *Rhamphorhynchus* would achieve a greater energy efficiency for flight in adults, although juvenile stages exhibited elevated metabolic rates due to rapid growth. On the other hand, *H. escuilliei*, with its semi-aquatic locomotion, would have adapted its metabolism for endurance rather than high bursts of energy, as seen in flight. Both groups demonstrated adaptations that reflect the need for metabolic capacity, albeit in different ways: while the aerobic energy costs of flight would be high, these would be offset by the flight efficiency in adult *Rhamphorhynchus* individuals, whereas the locomotion of *Halszkaraptor* would be more balanced, with lower energy expenditure during terrestrial and aquatic excursions.

## Caveats

Several species of the sample of extant birds used to construct the inference model utilized herein are characterized by the presence of pneumatization. The pneumatic foramina in long bones likely include arteries (in addition to the air sac), and this may have exerted effect on the size of the foramina and on the blood flow rate. As we needed an extant phylogenetic bracket to construct the inference model (*Witmer, 1995*), we need to include a sample of extant birds, assuming that the arteries passing through the pneumatic foramina are small and do not have a significant impact on the blood flow rate through the foramina.

## CONCLUSIONS

Our analyses highlight the aerobic capacity of the theropod *H. escuilliei* and the pterosaur *Rhamphorhynchus*. Despite significant differences in lifestyle and ontogenetic state, both taxa exhibited similar mass-independent MMRs and femoral nutrient artery blood flow

(Q̇). The mass-independent MMR of *H. escuilliei* aligned with that of the monitor lizard *V. gouldii* and certain birds emus and the Charadriiformes *Larus* and *Sterna*. However, the aerobic performance of *H. escuilliei* appears to fall below that of swimmers such as ducks and cursorial mammals. The juvenile *Rhamphorhynchus* analyzed herein likely exhibited elevated femoral blood flow when compared to mature individuals. While its endurance flight capabilities resemble those of migratory shorebirds, it is far from those seen in small migratory quails, *Coturnix*. In addition, adult *Rhamphorhynchus*, experiencing reduced growth-related demands, likely had lower mass-independent MMRs and operated under a more energetically efficient metabolic regime. Nevertheless, research is needed to clarify the ontogenetic influence on their metabolic demands and capacity.

## ACKNOWLEDGEMENTS

We are grateful to Professor Sophie Sanchez and ESRF Team for providing access to the micro-computed tomography (micro-CT) scan data and for her valuable feedback on an earlier version of this manuscript. We also sincerely thank Dr. Philipp Knaus and an anonymous reviewer for their constructive and insightful comments, which helped to improve the clarity and quality of this work.

### Funding

This work was supported by the European Research Executive Agency through the Marie Skłodowska-Curie Actions (MSCA) Postdoctoral Fellowship under the Project (FLAPS HORIZON-MSCA-2022-PF-01-01 no 101107135 to M. Sena) and the National Council for Scientific and Technological Development - CNPq (no 141138/2022-0 to E. Araujo). The funders had no role in study design, data collection and analysis, decision to publish, or preparation of the manuscript.

### Grant Disclosures

The following grant information was disclosed by the authors:
The European Research Executive Agency through the Marie Skłodowska-Curie Actions (MSCA) Postdoctoral Fellowship: no. 101107135.
The National Council for Scientific and Technological Development - CNPq: no. 141138/2022-0.

### Competing Interests

The authors declare there are no competing interests.

### Author Contributions

- Mariana V.A. Sena conceived and designed the experiments, performed the experiments, analyzed the data, prepared figures and/or tables, authored or reviewed drafts of the article, and approved the final draft.
- Dennis F.A.E. Voeten conceived and designed the experiments, analyzed the data, authored or reviewed drafts of the article, and approved the final draft.

- Esaú Araújo conceived and designed the experiments, analyzed the data, prepared figures and/or tables, authored or reviewed drafts of the article, and approved the final draft.
- Jorge Cubo conceived and designed the experiments, performed the experiments, analyzed the data, authored or reviewed drafts of the article, and approved the final draft.

## Data Availability

The data is available at GitHub and Zenodo:

- https://github.com/MAVAAS/MMR_Halszkaraptor_and_Rhamphorhynchus.git
- Mariana Sena. (2025). MAVAAS/MMR_Halszkaraptor_and_Rhamphorhynchus: Paleontology (Paleontology). Zenodo. https://doi.org/10.5281/zenodo.15830760.

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
