# Peer review of "Femoral metadiaphyseal and nutrient foramen perfusion suggests comparable maximal metabolic rates in a pterosaur and in a semi-aquatic maniraptoran dinosaur"

_PeerJ, doi:10.7717/peerj.19806_

## Round 0.1 · original submission · Minor Revisions

· Academic Editor

Minor Revisions

My apologies for the enormous delays in the review process here. Both referees have come back with minor revisions and I think the comments are generally fair and reasonable.

Reviewer 1 ·

Basic reporting

Sena et al. present a novel study on the inference of methabolic rates in the extinct pterosaur Rhamphorhynchus and the dromaeosaur Halszkaraptor using vascularization as a proxy to infer physiological adaptations. I really enjoyed reading the manuscript, which in my opinion, is well presented. I only have a few comments regarding some of the assumptions and implications reported in the manuscript: although implemantation of these comments is advised, I do not believe that they require major revision of the manuscript.

Experimental design

The experiments are well thought and elegant in their simplicity. I believe that these experiments will set a nice framework for future additional work using this proxy

The question is obviously important and well presented

The methods are clear and reproducible

Validity of the findings

The study is novel and impactful. I also appreciated how the authors integrated the limitations of the study in their discussion, which is, unfortunately, a rare sight.

Additional comments

- The authors suggest that the holotype of Halszkaraptor "likely reached maturity" (Line 193, Discussion). I believe that this is incorrect and I would argue that Halszkaraptor too is immature: in the original description (Cau et al. 2017 Nature), Extended Data Figures 4-5 show the presence of a single Line of Arrested Growth (LAG) across multiple skeletal elements. The lack of an External Fundamental System (EFS) and the scarsity of remodeling suggest that this animal was actively growing and far from being skeletally mature (sensu Griffin et al. 2021). Therefore, how is this changing your interpretation of the results? Can the authors expand more on the ontogenetic stage of Halszkaraptor based on available osteohisotlogical data?
- Correlation between ecomorphology and methabolic rates: the authors suggest that behavioral and ecological inferences can be drawn by the methabolic degree found in the two extinct taxa when compared with modern species with similar physiology. Can the authors justify this a bit more? Additionally, I want to remind the authors that alternative hypotheses regarding the ecology of Halszkaraptor exist and not necessarely support a wing propelled swimming habit (e.g. Brownstein 2019; Fabbri et al. 2022).
- I was kind of disappointed in seeing how a broad picture paragraph regarding the implications of this study on the evolution of archosaur methabolism was missing. In recent years, we have witnessed a broad variety of manuscripts aimed at resolving the evolution of methabolic rates across archosaurs (e.g. Chiarenza et al. 2024; Curry-Rogers et al. 2024; D'Emic et al. 2023; Grigg et al. 2022; Wiemann et al. 2022; Legendre & Devasne 2020). How do your results fit in these evolutionary models?

References:
- Brownstein, C. D. (2019). Halszkaraptor escuilliei and the evolution of the paravian bauplan. Scientific Reports, 9(1), 16455.
- Chiarenza, A. A., Cantalapiedra, J. L., Jones, L. A., Gamboa, S., Galván, S., Farnsworth, A. J., ... & Varela, S. (2024). Early Jurassic origin of avian endothermy and thermophysiological diversity in dinosaurs. Current Biology, 34(11), 2517-2527.
- Curry Rogers, K., Martínez, R. N., Colombi, C., Rogers, R. R., & Alcober, O. (2024). Osteohistological insight into the growth dynamics of early dinosaurs and their contemporaries. Plos one, 19(4), e0298242.
- D'Emic, M. D., O’Connor, P. M., Sombathy, R. S., Cerda, I., Pascucci, T. R., Varricchio, D., ... & Curry Rogers, K. A. (2023). Developmental strategies underlying gigantism and miniaturization in non-avialan theropod dinosaurs. Science, 379(6634), 811-814.
- Fabbri, M., Navalón, G., Benson, R. B., Pol, D., O’Connor, J., Bhullar, B. A. S., ... & Ibrahim, N. (2022). Subaqueous foraging among carnivorous dinosaurs. Nature, 603(7903), 852-857.
- Griffin, C. T., Stocker, M. R., Colleary, C., Stefanic, C. M., Lessner, E. J., Riegler, M., ... & Nesbitt, S. J. (2021). Assessing ontogenetic maturity in extinct saurian reptiles. Biological Reviews, 96(2), 470-525.
- Grigg, G., Nowack, J., Bicudo, J. E. P. W., Bal, N. C., Woodward, H. N., & Seymour, R. S. (2022). Whole‐body endothermy: ancient, homologous and widespread among the ancestors of mammals, birds and crocodylians. Biological Reviews, 97(2), 766-801.
- Legendre, L. J., & Davesne, D. (2020). The evolution of mechanisms involved in vertebrate endothermy. Philosophical Transactions of the Royal Society B, 375(1793), 20190136.
- Wiemann, J., Menéndez, I., Crawford, J. M., Fabbri, M., Gauthier, J. A., Hull, P. M., ... & Briggs, D. E. (2022). Fossil biomolecules reveal an avian metabolism in the ancestral dinosaur. Nature, 606(7914), 522-526.

·

Basic reporting

The English is professional altough not always unambiguous and clear. The manuscript could be improved by a final review by a native speaker. Literatures references are appropriate and give sufficient field background and context. The Article structure is professional, although the result section is a bit short. Putting the recovered data into the context of the major groups of recent taxa might make it a bit longer. Raw data is completely shared through a link at the end of the article. Figures are accurate, altough the 3D scan of the second bone in figure 1 should be shown in completion to show the location of the nutrient canal on the bone. The article is self-contained with relevant results to the hypotheses provided.

Experimental design

The article presents primary research. The research question could be defined more clearly. It is very interesting, that the study for the first time combines more accurate ways of estimating blood flow in extinct animals. This could be made more clear in the introduction and how it thus is important to elucidating the unsual niches hypothsized for the two studied species. The investigation is performed to a high technical and ethical standard. The methods are described with large detail although some wording in the methods section is slightly ambiguous and could be made more precise. See details in the commented manuscript. The R script and raw data provided allow to replicate the software analysis section of the study.

Validity of the findings

Altough the underlying synchrotron scans are not provided, these files are usually so enormous in size that it is not practical as of yet to provide a link. However, surface wavefront files or smaller volume files such as image stacks might be feasable to provide online. It would be best practice and I would recommend to provide these files if possible. The statistical data provided are robust, statistically sound and controlled. The conclusions are well stated, linked to original resarch question and limited to supporting results.

Additional comments

Overall a very interesting article. Although the wording and formatting needs to be improved in several instances, the results are interesting and well worth publishing. Also applying the new formulae to extinct taxa is very interesting and promising in the future.

---

## Round 0.2 · accepted · Accept

· Academic Editor

Accept

Thank you for making the changes and updates to the work.

It would be crucial that the data/script stored on Github is linked with a resource offering a permanent object identifier (e.g., Zenodo) so please attend to this as soon as possible.